# Delayed increase in stone tool cutting-edge productivity at the Middle-Upper Paleolithic transition in southern Jordan

Seiji Kadowaki [1] ✉, Joe Yuichiro Wakano[2], Toru Tamura[3], Ayami Watanabe[1], Masato Hirose [4], Eiki Suga[5], Kazuhiro Tsukada[1], Oday Tarawneh[6] & Sate Massadeh[7]

Although the lithic cutting-edge productivity has long been recognized as a quantifiable aspect of prehistoric human technological evolution, there remains uncertainty how the productivity changed during the Middle-to-Upper Paleolithic transition. Here we present the cutting-edge productivity of eight lithic assemblages in the eastern Mediterranean region that represent a chrono-cultural sequence including the Late Middle Paleolithic, Initial Upper Paleolithic, the Early Upper Paleolithic, and the Epipaleolithic. The results show that a major increase in the cutting-edge productivity does not coincide with the conventional Middle-Upper Paleolithic boundary characterized by the increase in blades in the Initial Upper Paleolithic, but it occurs later in association with the development of bladelet technology in the Early Upper Paleolithic. Given increasing discussions on the complexity of Middle-Upper Paleolithic cultural changes, it may be fruitful to have a long-term perspective and employ consistent criteria for diachronic comparisons to make objective assessment of how cultural changes proceeded across conventional chrono-cultural boundaries.

The Middle to Upper Paleolithic transition (the MP-UP transition) has long been known as a key prehistoric cultural process or chronological range to investigate human biocultural evolution due to its temporal proximity to the wide geographic expansion of *Homo sapiens* ca. 50–40 thousand years ago and concomitant demise of archaic humans including Neanderthals[1-3]. Mainly based on European records, the MP-UP cultural transition was conventionally regarded as a discontinuous process or "revolution" marked by the introduction of new cultural/behavioral packages associated with a wave of dispersing *Homo sapiens* population[4,5]. However, more recent increase in the recovery of archeological records, particularly in Africa and Asia, has suggested

geographically diverse cultural patterns that involve both continuous and discontinuous aspects as well as various timings of change depending on cultural elements[6-9].

The cultural framework of the MP-UP transition has been effectively delineated by techno-morphological attributes of stone tools, which constitute the most abundant cultural remains at Paleolithic sites[10-12], and detailed examinations of lithic techno-morphology led researchers to reassess the conventional view and propose more complicated cultural processes at the MP-UP transition[2,13-15]. However, in order to compare variations of lithic assemblages over different periods or regions, it remains a methodological challenge for archeologists to establish consistent and quantitative criteria (rather than

[1]Nagoya University Museum, Nagoya University, Furo-cho, Chikusa-ku, Nagoya 464-8601, Japan. [2]School of Interdisciplinary Mathematical Sciences, Meiji University, Nakano 4-21-1, Nakano-ku, Tokyo 164-8525, Japan. [3]Geological Survey of Japan, AIST, Central 7, 1-1-1 Higashi, Tsukuba, Ibaraki 305-8567, Japan. [4]Laboratory of Archaeology, Kiso Regional Union, Nagano 399-6101, Japan. [5]Graduate School of Environmental Studies, Nagoya University, Furo-cho, Chikusa-ku, Nagoya 464-8601, Japan. [6]Department of Antiquities, Third Circle, Jabal Amman, Amman, Jordan. [7]Ministry of Tourism and Antiquities, Third Circle, Jabal Amman, Amman, Jordan. ✉e-mail: kadowaki@num.nagoya-u.ac.jp

descriptions) that allow objective illustration and assessment of temporal/spatial cultural dynamics.

Here, we present a quantitative examination of diachronic changes in stone tool assemblages at the MP-UP transition by focusing on the production rate of stone tool cutting edges. Most Paleolithic stone artifacts have sharp edges as their functional parts (e.g., for cutting and scraping), and sharp edges are created by knapping off flakes from silicious rocks, such as obsidian and flint[16]. Thus, the length of cutting edge per mass of stone has been used in numerous studies to quantify the efficiency in stone tool production, in other words, the efficiency in the consumption of raw rock material[17–22]. Although the cutting-edge production rate has been widely recognized as a consistent and quantifiable aspect of prehistoric human technological evolution, there remains uncertainty about how the rate actually changed during the MP-UP transition. Conventionally, a shift from dominant Levallois technology in the MP to the blade technology in the UP was considered to have increased the cutting-edge production rates[23,24]. However, uncritical recourse to such a progressive view is currently being reconsidered under recent recognition of Paleolithic technological variability and insights from lithic experimental studies (See Supplementary Note 1 for details).

In addition, it is well known that lithic production activities can vary depending on the availability of raw materials, and particularly in the case of Paleolithic mobile foragers, tool production and use activities are often segmented by residential moves and performed in multiple stages over different places[25–27]. This may cause synchronic variability in lithic assemblages depending on the nature of sites, such as raw material quarry/workshop, short-term transitory site, and base camp. Because our main concern in this study is diachronic changes rather than synchronic variations, it is critical to use research materials/lithic

assemblages that share the same raw material condition and occupational nature. To this end, we selectively used eight assemblages from five sites in the Levant (east Mediterranean region), focusing on southern Jordan, where the sites share the same environmental settings, particularly in terms of the availability of lithic raw material (Fig. 1; Methods; Supplementary Figs. 1–5). All the five sites are situated under rock shelters and represent small habitation sites with hearth remains, where palimpsest of lithic production and use activities accumulated[28–31].

Importantly, the selected lithic assemblages cover the MP-UP transition (Supplementary Fig. 6), including two assemblages of the Late Middle Paleolithic (LMP) from Tor Sabiha and Tor Faraj, two Initial Upper Paleolithic (IUP) assemblages from Wadi Aghar and Tor Fawaz, two Early Upper Paleolithic (EUP) assemblages from Tor Hamar Layers H and Layers F–G, and two Epipaleolithic (Epipal) assemblages from Tor Hamar Layer E2 and Layers B–E1 (Supplementary Table 1). The two assemblages in each of the EUP and the Epipal are stratigraphically sequenced at Tor Hamar, but we do not assume a chronological relationship between the two assemblages within each of the LMP and IUP. Thus, chrono-cultural units assumed in this study are the LMP (Tor Sabiha and Tor Faraj), IUP (Wadi Aghar and Tor Fawaz), EUP1 (Tor Hamar Layers H), EUP 2 (Tor Hamar Layers F–G), Early Epipal (Tor Hamar Layer E2), and Middle Epipal (Tor Hamar Layers B–E1).

Regarding the MP-UP transition in the Levant, a slight increase in the rate was previously detected by two studies that actually measured the cutting-edge length of archeological specimens, but the early studies were based on small sample size, lacked a detailed chrono-cultural sequence from the IUP to EUP, and employed unprecise methods for edge-length measurement[19,21,29]. This paper updates the previous studies by adding IUP and EUP (more specifically, the Ahmarian) assemblages that establish more detailed cultural

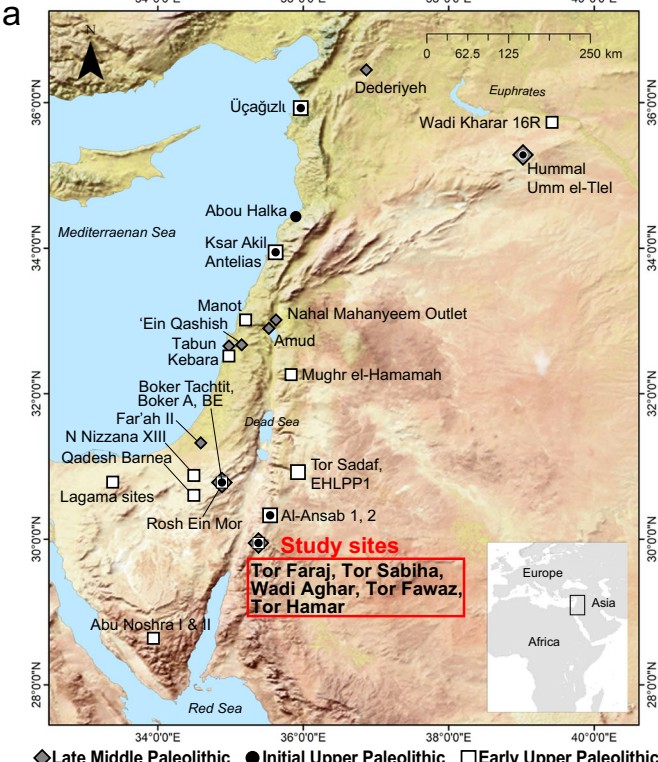

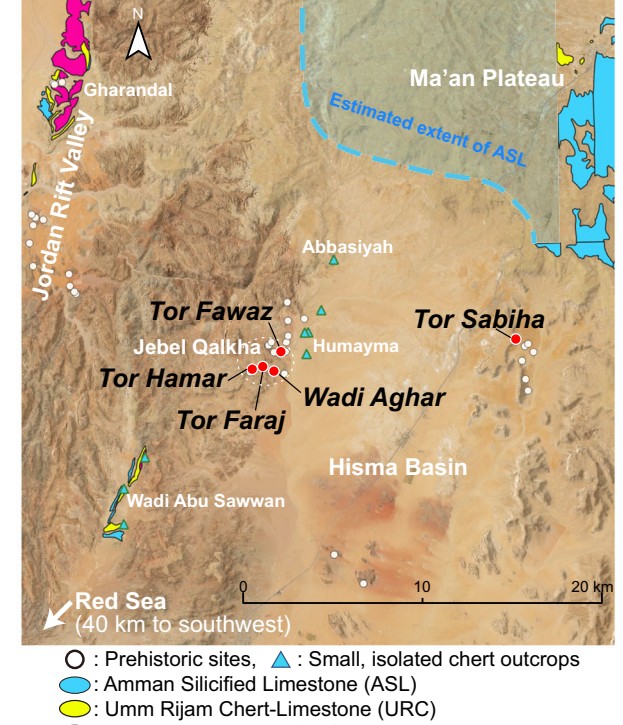

◆ Late Middle Paleolithic  ● Initial Upper Paleolithic  ☐ Early Upper Paleolithic

○ : Prehistoric sites,  △ : Small, isolated chert outcrops
⬤ : Amman Silicified Limestone (ASL)
⬤ : Umm Rijam Chert-Limestone (URC)
⬤ : Dana Conglomerate (DC)

**Fig. 1 | The location of study sites (Tor Faraj, Tor Sabiha, Wadi Aghar, Tor Fawaz, and Tor Hamar) in southern Jordan. a** Map of the Levant showing Late Middle Paleolithic, Initial Upper Paleolithic, and Early Upper Paleolithic sites. Early Upper Paleolithic sites include those with Ahmarian and similar bladelet assemblages. The maps were made with Natural Earth. **b** Satellite image (OpenStreetMap)

of the western Hisma Basin in southern Jordan, showing the four study sites and the distributions of chert outcrops according to geological maps[87,97] and survey[30]. Data of the prehistoric sites are from refs. 29,98. Note that chert outcrops in the Hisma Basin are small and sporadic, and such raw material settings are shared by the five study sites.

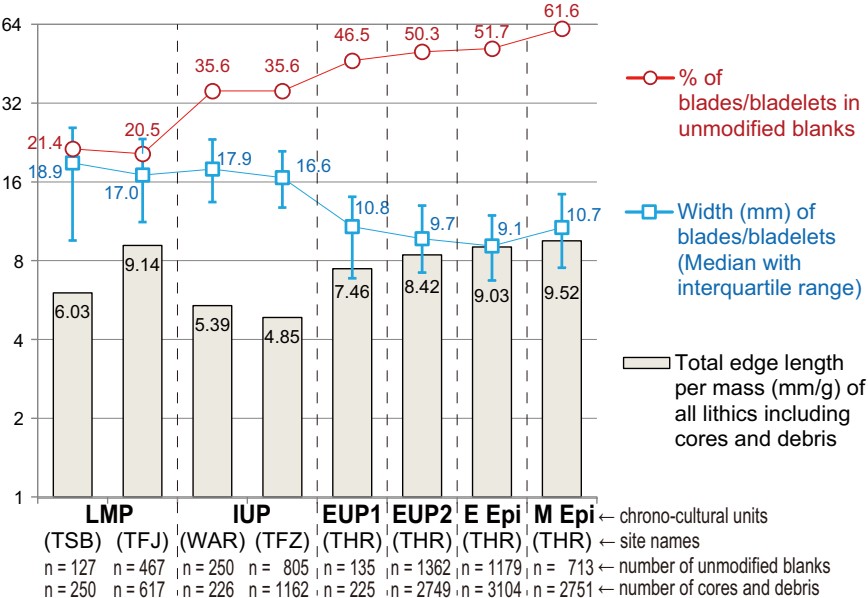

**Fig. 2 | Diachronic changes in the total edge length per mass (mm/g) of all lithics along with the width (mm) of blades/bladelets and the percentage of blades/bladelets in unmodified blanks (excluding CTEs and spalls).** Although the compared assemblages are aligned in chronological order from left to right, we do not assume a chronological relationship between LMP (TSB) and LMP (TFJ) and between IUP (WAR) and IUP (TFZ), as described in the text. LMP, Late Middle Paleolithic; IUP, Initial Upper Paleolithic; EUP, Early Upper Paleolithic; E Epi, Early Epipaleolithic; M Epi, Middle Epipaleolithic. The lithic assemblages were excavated from Tor Sabiha (TSB), Tor Faraj (TFJ), Wadi Aghar (WAR), Tor Fawaz (TFZ), and Tor Hamar (THR) located in southern Jordan. CTE core trimming element. Source data are provided as a Source Data file.

sequences at the MP-UP transition. The addition of the IUP assemblages is particularly important because this chrono-cultural concept is used not only in the Levant but also in other regions, such as Central–Southeastern Europe and Central–North Asia, to characterize unique cultural changes at the beginning of the UP (Supplementary Note 2)[2,14,32,33]. The IUP and the Ahmarian lithic technology have also been increasingly recognized as key cultural records that can be correlated to paleoanthropological and genetic evidence to examine the dispersal processes of *Homo sapiens* in Eurasia[2,34,35].

To make an accurate assessment of cutting-edge productivity, we measured the cutting-edge length of all debitage types except for retouched pieces, cores, and debris (such as chips and chunks), following methods proposed by recent lithic experimental studies[18,20]. To achieve accurate and precise measurement of irregular edge forms, a rigorous protocol of digital measurement was applied to the cutting-edge length of more than 5000 stone tools[18,20] (Methods; Supplementary Fig. 7).

The following results show that a major increase in cutting-edge productivity does not coincide with the conventional MP-UP boundary characterized by the increase in blades in the IUP, but it occurs later in association with the development of bladelet technology in the EUP. The low cutting-edge productivity in the IUP derives from the large mass and volume of blanks with wide striking platforms. The degree of the productivity increase in the EUP, realized by the miniaturization of multiple blank types, is greater than the subsequent one in the Epipal.

## Results

### Temporal trends in the cutting-edge length per mass from LMP to Epipal

From the LMP to the IUP, the relative frequencies of blades/bladelets clearly increased without significant changes in their size (Fig. 2). In contrast, from the IUP to EUP, blades/bladelets increased only slightly, but their size clearly decreased, reflecting the development of bladelet technology. From the EUP to the Early Epipal, the relative frequency of blades/bladelets did not change significantly, and their size reduction was marginal in comparison with that of the IUP-EUP

transition. From the Early to Middle Epipal, the relative frequency and size of blades/bladelets increased.

Regarding the total cutting-edge length/mass ratio, the two IUP assemblages (Wadi Aghar and Tor Fawaz) showed the lowest values among the lithic assemblages analyzed in this study. Diachronically, the total length/mass ratio decreased from the LMP (particularly from Tor Faraj) to the IUP and then distinctively increased in the EUP, followed by a subtle increase in the Epipal.

A similar temporal trend was observed in the individual length/mass ratios (Fig. 3a), which represent the cutting-edge length per mass for each of the unmodified blanks (Methods). Pairwise comparisons by the Dunn–Bonferroni test detected significant differences between 21 pairs out of all possible pairs (=28) among the eight assemblages. All of the 21 significant differences are consistent with a proposed diachronic change through which the edge length/mass ratio decreased from the LMP to IUP, increased from the IUP to EUP, and again increased from the EUP to Epipal. The data of the LMP (Tor Sabiha) is not statistically different from those of the two IUP assemblages, which are the only differences that are expected by the proposed diachronic change but not statistically significant.

### Variations in the cutting-edge length/mass ratio by debitage types

Clearly, bladelets show the greatest ratios among the debitage types in all periods from the LMP to Epipal (Fig. 3b). In more detail, however, the length/mass ratios of bladelets from the two IUP sites are lower than those of the LMP, EUP, and Epipal assemblages ($p < 0.03$ by Dunn–Bonferroni test for the differences between IUP and EUP/Epipal). Similarly, flakes and blades of the two IUP sites tend to show lower length/mass ratios than the LMP, EUP, and Epipal (see Supplementary Tables 2–4 for statistical significance).

### Relations of the cutting-edge length/mass ratios to dimensional attributes of blanks

Clear negative correlations were observed between the length/mass ratio and the mass of each lithic artifact (Supplementary Fig. 8),

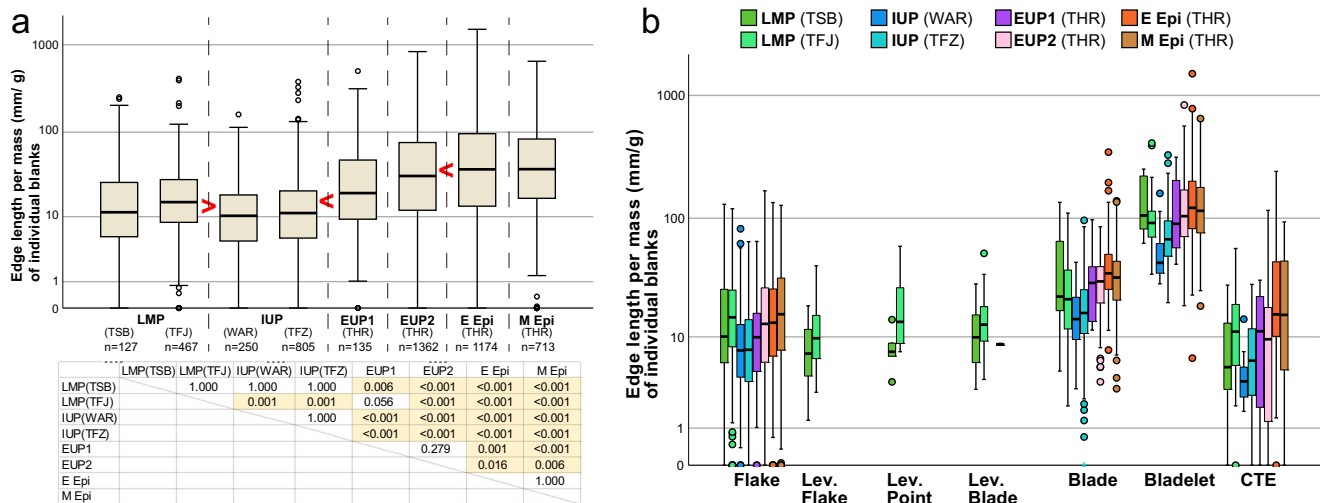

**Fig. 3 | Edge length per mass (mm/g) of individual blanks by lithic assemblages and by debitage types.** The box plots indicate the median (middle line) and interquartile range (box). The upper and lower ends of the whiskers represent the maximum and minimum, excluding outliers (single points) defined by the Tukey method. **a** Diachronic changes in the edge length/mass ratios of individual blanks according to the eight assemblages from LMP to Epipal (see Fig. 2 for the abbreviations of period- and site-names). Although the compared assemblages are aligned in chronological order from left to right, we do not assume a chronological relationship between LMP (TSB) and LMP (TFJ) and between IUP (WAR) and IUP

(TFZ), as described in the text. Inequality symbols (> and <) in box plots note statistically significant differences only for box plots next to each other. The table shows the *p*-values of the pairwise comparisons by the Dunn–Bonferroni test (two-sided). The values lower than 0.05 are highlighted. **b** Edge length/mass ratios of individual blanks according to debitage types and assemblages (see Supplementary Table 1 for *n* values). See Supplementary Tables 2–4 for statistical significance of differences in flakes, blades, and bladelets among the assemblages. Lev. Levallois, CTE core trimming element. Source Data are provided in the figshare. https://doi.org/10.6084/m9.figshare.23577093.

although this result is self-evident because the mass is used in the calculation of the length/mass ratio. More importantly, other attributes that showed clear negative correlations with the edge length/mass ratio are width, thickness, and the platform area, as indicated by high absolute values of Spearman's Rs (Figs. 4–5; Supplementary Figs. 9 and 10). This means that narrower and thinner blanks with smaller striking platforms tend to have greater ratios of edge length per mass. As typically shown in the case of width (Fig. 4), such a negative correlation is observable similarly in each of the debitage types. Weaker correlations were observed for length, the ratio of length to width (i.e., elongatedness), and the ratio of width to thickness (i.e., flatness) (Supplementary Figs. 11–13). These patterns were consistently observed in all the assemblages from the LMP to Epipal, and no diachronic trend was observable (Fig. 5).

## Discussion

The first type of the edge production rate, "the total edge length/mass ratio", is theoretically close to the ratio of cutting-edge to core mass (CE:CM) in Eren et al.[18] and the cutting edge per gram of core (mm/g) in Muller and Clarkson[20], both of which are experimental lithic production studies. In its application to archeological materials, we need to consider the possibility that the length/mass ratio can be affected by factors other than the edge production efficiency because archeological lithic remains are palimpsests of multiple different activities, including not only lithic productions from several raw material units but also selective import or removal of some lithic artifacts for the purpose of their future use or cleaning of activity areas. For example, the total edge length/mass ratio may be underestimated if the archeological lithic assemblage was collected from a refuse dump where cores and chunks were concentrated. Cores and chunks do not provide sharp cutting edges, but their mass lowers the total length/mass ratio. In contrast, the total edge length/mass ratio may be overestimated if the lithic assemblage was collected from sites or activity areas where usable blanks were selectively imported.

To consider the above possible influences, we plotted the total length/mass ratio against the mass ratio of cores and debris in each of

the lithic assemblages (Fig. 6). Theoretically, a negative correlation is expected between these two factors because the smaller mass ratio of cores and debris should contribute to the increase in the total edge length/mass ratio. This factor may explain the IUP's lower ratios of edge length per mass than the LMP because the IUP assemblages have a higher mass ratio of cores and debris than the LMP. On the other hand, the two IUP assemblages have lower ratios of edge length per mass than the EUP and Epipal assemblages despite the IUP's smaller mass ratios of cores and debris than the EUP.

The low efficiency of cutting-edge production in the two IUP sites was also suggested by their low values of length/mass ratios of individual pieces (Fig. 3a), which only consider unmodified blanks and do not include a mass of cores and debris in calculations. This indicates that some morphological characteristics of unmodified blanks contributed to the decrease in edge length/mass ratios of the IUP. One such factor is the frequency of bladelets that show the greatest ratios of edge length per mass among the debitage types regardless of the time periods (Fig. 3b). The low occurrences of bladelets in the IUP (Supplementary Table 1) or the large size of blades/bladelets (Fig. 2), partly explain the low ratios of edge length per mass in the IUP assemblages.

In addition, we suggest that other morphological factors are width, thickness, and the size of the striking platform, which showed strong negative correlations with the edge length/mass ratios (Fig. 5). These correlations are consistent with the results of a lithic experimental study by Muller and Clarkson[20]. In our archeological materials, unmodified blanks of the IUP assemblages are significantly wider and thicker than those of the EUP and Epipal assemblages, and the striking platforms are also larger in the IUP (Supplementary Figs. 14–16). These morphological characteristics transcend the debitage types because the two IUP assemblages show low edge-length/mass ratios in multiple debitage types, including flakes, blades, and bladelets (Fig. 3b).

In this way, the results of this study suggest that the cutting-edge productivity of the IUP was closer to the LMP than to the EUP (see Supplementary Discussion for a more detailed comparison between the LMP and IUP). This does not fit the conventional affiliation of the

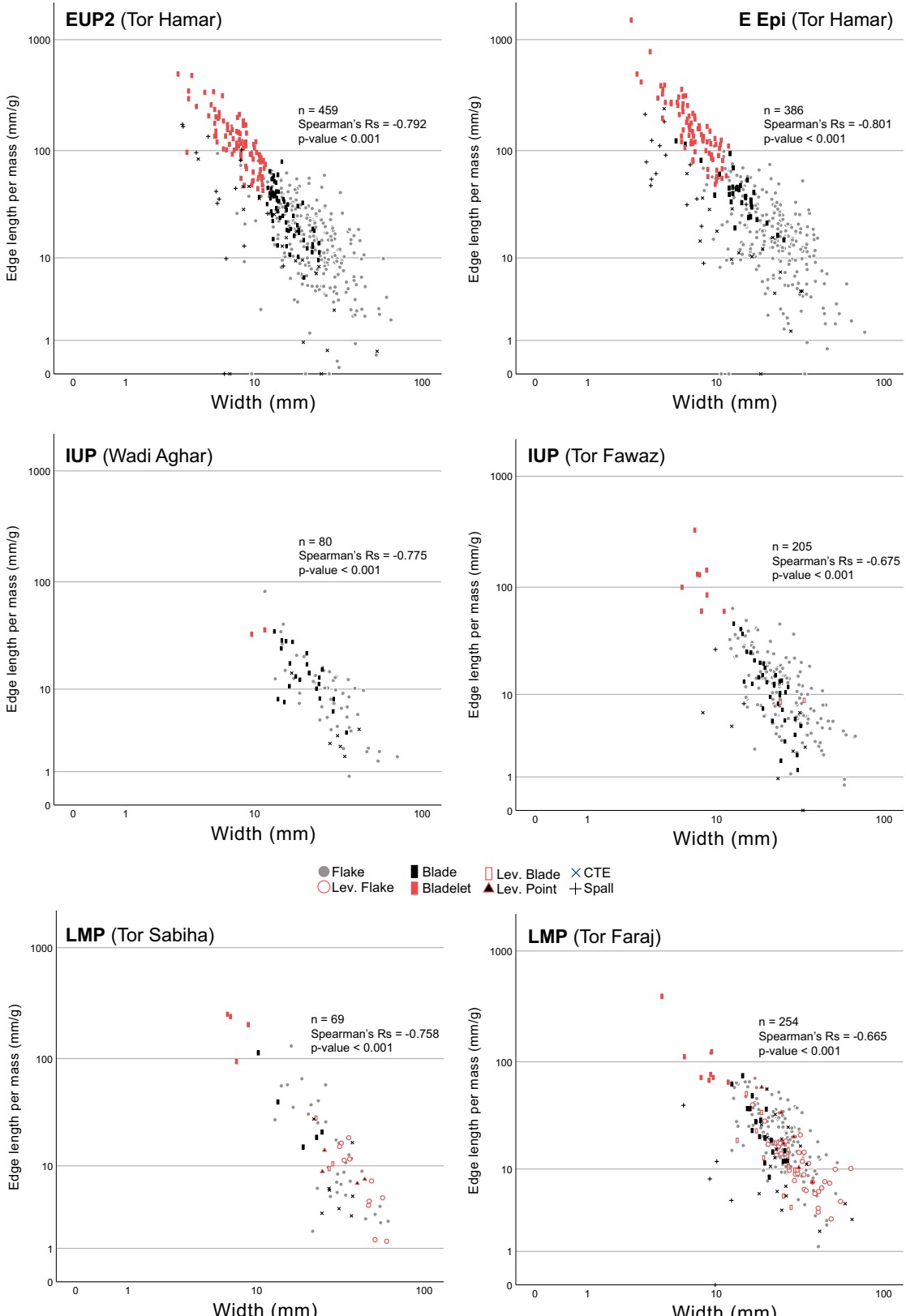

**Fig. 4 | Scatterplots between the edge length per mass (mm/g) of complete lithic specimens and their width.** LMP Late Middle Paleolithic, IUP Initial Upper Paleolithic, EUP Early Upper Paleolithic, E Epi Early Epipaleolithic. The plots include different symbols according to debitage types. Lev. Levallois, CTE core trimming element. The significance testing of the Spearman correlation coefficient was two-sided. Source Data are provided in the figshare. https://doi.org/10.6084/m9.figshare.23577093.

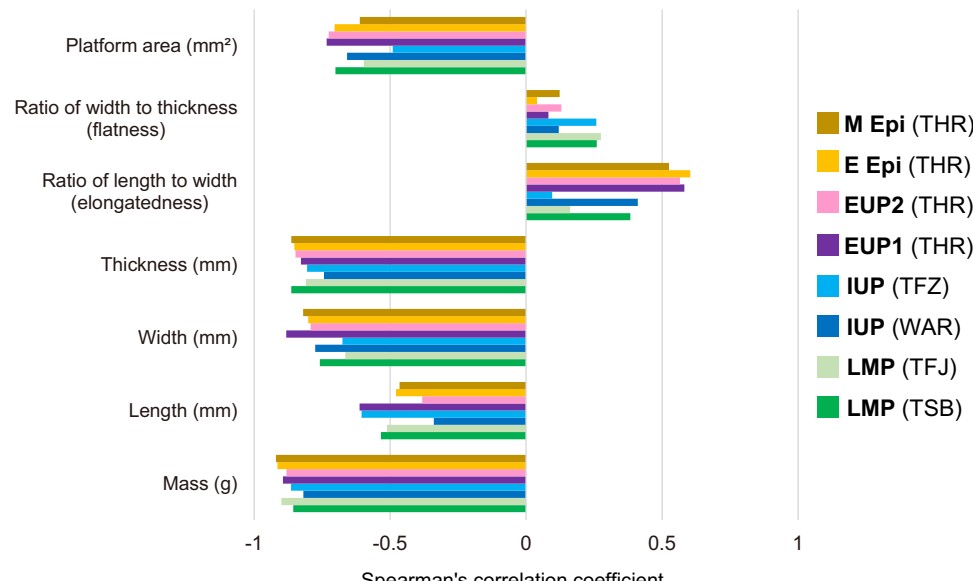

**Fig. 5 | Spearman's Rs values for the correlations between the edge length per mass (mm/g) of complete lithic specimens and their various morphological measurements.** LMP Late Middle Paleolithic, IUP Initial Upper Paleolithic, EUP Early Upper Paleolithic, E Epi Early Epipaleolithic, M Epi Middle Epipaleolithic. The lithic assemblages were excavated from Tor Sabiha (TSB), Tor Faraj (TFJ), Wadi Aghar (WAR), Tor Fawaz (TFZ), and Tor Hamar (THR) located in southern Jordan. Source data are provided as a Source Data file.

IUP within the UP category and implies a complexity of the cultural process (rather than a revolution) at the MP-UP transition.

Regarding the behavioral significance of the cutting-edge production rate, it was not the only lithic technological behavior related to the adaptive advantage of prehistoric hominins, but it was only a part of lithic technological organization consisting of multiple behaviors, such as the acquisition of lithic raw material, tool production, use, maintenance, transport, re-use, recycling, and discard, which were further linked to other activities like subsistence and mobility[21,22,27]. This means that we have to be cautious about simply linking cutting-edge productivity to the gross evaluation of "technological progress". Instead, the low cutting-edge productivity can be a result of technological choice reasonably linked to other technological behaviors that provide adaptive advantages in certain environments.

As discussed above, the low cutting-edge productivity in the IUP derives from the large mass and volume of blanks. Several lithic studies have already suggested that large blanks have the potential for prolonged use through retouching or re-sharpening of their cutting edges so that the total edge length throughout the tool's use-life can accumulate[18,22,36]. Ethnoarchaeological and archeological studies showed that such curated tools were often carried with mobile foragers and functioned as personal tool kits[22,37] (or the provisioning of individuals[38]) so that they reduced the task and risk in the lithic raw material procurement and tool production between frequent residential movements particularly where raw material availability was unpredictable[39].

In the Levantine IUP, robust (often pointed) blanks are known to have been retouched or curated in several different ways, resulting in a range of retouched tool types (such as end scrapers, burins, chamfered pieces, and several point types)[11,15,40,41] as well as cores (such as cores-on-flake and burin-cores)[42]. For example, lithic assemblages from Layers I–C at Üçağızlı suggested the transport of finished tools and/or large blanks from distant flint sources (15–30 km away)[43], and end scrapers from these layers were found to be more reduced than those from upper layers (Ahmarian). Importantly, this strategy, i.e., provisioning individuals, was associated with frequent residential moves and brief occupations that were indicated by small and discrete distributions of hearths and other cultural remains as well as by greater reliance on high-ranked food resources[44].

Such an ephemeral nature of IUP occupation is also indicated by a low density of cultural remains and a narrow range of on-site activities at Wadi Aghar[45] analyzed in this study. Surface scatters or redeposition of IUP artifacts, often mixed with LMP or EUP remains at many sites, likely resulted from the erosions of ephemeral IUP occupational deposits[46–49]. Increased mobility during the IUP is also suggested by a range expansion of resource procurement, as indicated by the import of marine shells to inland IUP sites at Wadi Aghar and Tor Fawaz in southern Jordan[45,50]. Even if the marine shells could have been indirectly obtained from other groups, such intergroup exchanges still likely involved movements of people for interactions. However, a few IUP sites indicate more substantial occupations associated with intensive lithic production from imported chert nodules (e.g., Boker Tachtit[51,52] and Tor Fawaz[42,53]). Thus, more investigations are needed to clarify the variability of IUP occupations and to understand how the production and use of IUP tools and cores were organized in relation to subsistence activities and residential movements.

Ephemeral occupations are also known in the LMP, particularly at open-air sites in the eastern Mediterranean region like 'Ein Qashish[54], Far'ah II[55], Hummal[56], and Nahal Mahanayeem Outlet[57], but more intensive occupations have been suggested for other LMP sites, particularly cave and rock shelter sites, which show dense and rapid depositions of cultural remains[58–60]. In association with these diverse settlement patterns, mixed strategies for stone-tool provisioning were employed in the LMP, including the import of chert nodules or initially prepared cores from local and non-local sources[59] as well as the selective transport of Levallois blanks and retouched tools[61].

In this study, the Tor Faraj LMP assemblage showed a relatively high rate of cutting-edge production in comparison with the IUP and another LMP assemblage (Tor Sabiha). A series of previous studies of lithic assemblages at Tor Faraj suggested the bulk import of chert nodules and their intensive core reduction at site[59], which served as a base camp associated with spatially organized hearths and structured use of space. Such intensive and recurrent occupations at Tor Faraj contrast to more transitory occupations with sparse cultural remains at Wadi Aghar (IUP)[45] and Tor Sabiha (LMP)[29], where cutting-edge

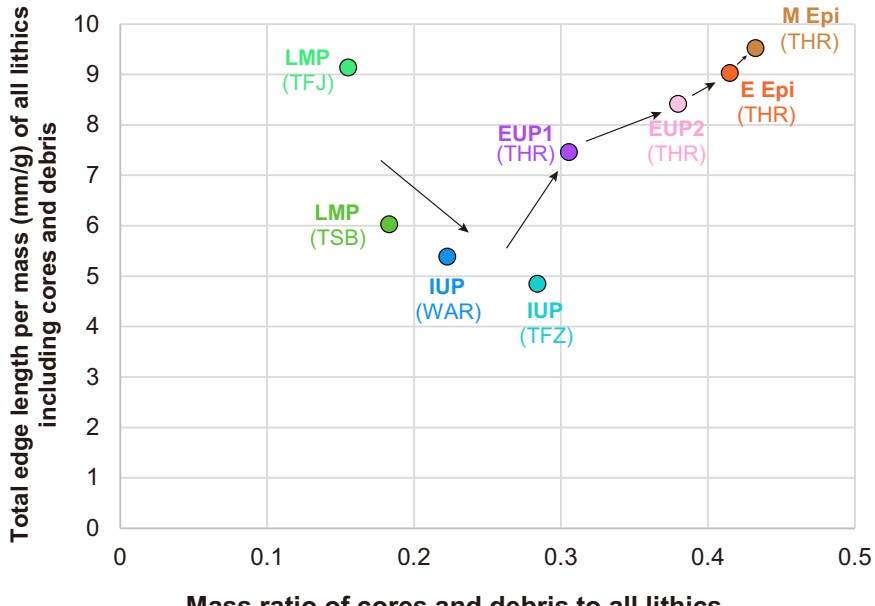

**Fig. 6 | Correlations between the total edge length/mass ratio and the mass proportion of cores and debris in the lithic assemblages.** Theoretically, the ratio of total edge length/mass (mm/g) is expected to decrease as the mass ratio of cores and debris increases. Such a negative correlation is observable only in LMP and IUP. Arrows indicate a chronological order of the lithic assemblages. Note that we do not assume a chronological relationship between LMP (TSB) and LMP (TFJ) and between IUP (WAR) and IUP (TFZ), as described in the text. LMP Late Middle Paleolithic, IUP Initial Upper Paleolithic, EUP Early Upper Paleolithic, E Epi Early Epipaleolithic, M Epi Middle Epipaleolithic. The lithic assemblages were excavated from Tor Sabiha (TSB), Tor Faraj (TFJ), Wadi Aghar (WAR), Tor Fawaz (TFZ), and Tor Hamar (THR) located in southern Jordan. Source data are provided as a Source Data file.

productivity was low. However, this explanation is not sufficient for the low rate of cutting-edge production at Tor Fawaz, where densely distributed lithic artifacts indicate abundant lithic production activities from imported chert cobbles.

From the IUP to EUP (Ahmarian), our Jordanian assemblages showed a significant increase in cutting-edge productivity, and the degree of this increase was even greater than a subsequent one from the EUP to Epipal. The edge production increase in the EUP resulted from the reduction in blanks' mass, width, thickness, and size of striking platforms according to their negative correlations with cutting-edge productivity (Fig. 5). Such miniaturization in blanks is consistent with prevailing views about the development of bladelet technology in the EUP, specifically the Ahmarian industry[14,62,63].

In addition, our analyses demonstrated that the size reduction occurred not only in blades/bladelets but also in flakes (Supplementary Figs. 14–17), and the accumulating effects of multiple blank types led to an overall significant increase in the assemblage's cutting-edge productivity. We admit the functional advantages of bladelets as part of projectile weapons and composite tools[64–66], but the percentage of bladelet points (such as el-Wad points) in bladelet blanks is usually very low, e.g., about 5% in Ahmarian assemblages in southern Jordan[62]. These observations suggest that bladelets were not the sole factor that characterizes the IUP-EUP transition, but another major technological change was the increased efficiency in raw material consumption realized by the miniaturization of multiple blank types. Although the Epipal is characterized by the increase in various forms of microliths created by abrupt retouching, our analysis supports the idea that significant miniaturization of blanks occurred earlier in the EUP[65].

The increased productivity of cutting edges in the EUP and Epipal was accompanied by the diversification of mobility patterns and lithic provisioning strategies. Like the IUP, many EUP and Epipal sites are small, consisting of a few hearths, and most likely represent transitory camp remains by mobile hunter-gatherers[67,68]. However, some UP sites indicate intensive occupations particularly at cave sites[69], and some large aggregation settlements appear in the Epipal[70]. For example, UP

occupations at Üçağızlı became more intensive during the EUP (Ahmarian), as indicated by expanded dietary breadth and greater density of stones, bones, and ash in comparison with the IUP occupations[43]. Along with this change in residential behavior, the lithic provisioning strategy at Üçağızlı shifted from the IUP emphasizing the curation of portable tool kits to the EUP employing the import of nodules or partially prepared cores in bulk from distant sources of good quality chert, i.e., the provisioning of places[43]. In the latter system, the increased productivity of cutting edges, realized by lithic miniaturization, was beneficial as it enabled the economical consumption of raw materials stored at base camps.

Our previous studies of the lithic assemblages in southern Jordan also showed a major change in the use of lithic raw material, not between the LMP and the IUP but between the IUP and the EUP (Supplementary Fig. 18)[31,71]. Importantly, a raw material type that increased in the EUP (Type FH: fine-grained and translucent chert) was more frequently used for the production of small blanks, including bladelets, than another chert type (Type M: medium-grained chert) that was more dominant in the LMP and IUP[71,72]. This is consistent with the results of this study, indicating that a major change in the raw material economy did not necessarily coincide with the classic MP-UP boundary marked by the increase in blades but occurred in tandem with lithic miniaturization that occurred in the EUP and then continued to the Epipal.

Lastly, our results and discussions can serve as a working hypothesis relevant to wide geographic regions, including not only the Levant and West Asia but also Europe and Central-North Asia, where similar IUP-EUP lithic technological changes have been observed. The Levantine IUP lithic assemblages share basic techno-typological elements and chronological positions with those in Europe and Central–North Asia[2,73–75], and their occurrences were associated with the geographic expansion of *Homo sapiens* to those areas[35,73]. The subsequent EUP/Ahmarian industry is characterized by the development of bladelet technology, whose geographic spread in Europe has also been discussed in relation to the subsequent dispersal of *Homo*

*sapiens* from the Levant[2,4,34]. More broadly, our discussions on the increased cutting-edge productivity in the EUP in tandem with the lithic miniaturization and the changes in raw material provisioning/selection may be analogous to those regarding the lithic miniaturization process and its behavioral implications at the Middle to Later Stone Age transition in Africa[76–78].

In conclusion, the results of this study showed that a major increase in cutting-edge productivity did not coincide with the conventional MP-UP boundary characterized by the increase in blades in the IUP, but it occurred later in association with the development of bladelet technology in the EUP. However, this study does not negate the differences between LMP and IUP in other cultural aspects, such as core reduction technology, tool typology, hunting tools, organic implements, and symbolic behaviors[10,14,41,44,52,79,80]. The low productivity of cutting-edge in the IUP could have been part of its unique technological strategies that focused on the provisioning of individuals with large lithic blanks as portable tool kits for prolonged use. However, more studies are necessary to clarify the variability in mobility and lithic technological organization of the IUP and their differences from the LMP or EUP.

Consequently, we suggest that the MP-UP cultural transition was not a single sudden replacement, but it should be regarded as a more complicated evolutionary process involving multiple aspects, and their changes occurred over a long time at various times. A certain degree of graduality in the MP-UP cultural transition has been recognized in some of IUP techno-morphological features, such as typological Levallois points, surficial core exploitation, platform faceting, and hard hammer percussion[13,32,41,52]. However, the complexity of cultural change was rarely examined through quantitative comparisons over the long-term, partly due to unstandardized systematics of lithic artifacts among different time periods. To overcome this methodological problem, this study focused on the cutting-edge production rates as a consistent measure, and the long-term diachronic examinations over the LMP, IUP, EUP, and Epipal allowed us to highlight a similarity between the LMP and IUP and a major change from the IUP to EUP in the cutting-edge production rates. Given increasing discussions on the complexity of MP-UP cultural changes[81], it may be fruitful to have a long-term perspective to make an objective assessment of how cultural changes proceeded across conventional (artificial) chronological boundaries.

## Methods
### Paleolithic assemblages in southern Jordan analyzed in this study
For the study of cutting-edge production rates, we used eight Paleolithic assemblages from five sites in the western Hisma Basin, southern Jordan. Our archeological fieldwork in Jordan was conducted according to the "Regulations for Archeological Projects in Jordan based on the provisions of the Jordanian Antiquities Law Number 21 for the year 1988 and its amendments". The permissions to conduct the fieldwork and the export of archeological materials to Japan, where analyses were conducted, were obtained from the Department of Antiquities of Jordan.

The following is the list of permissions for our fieldwork and the export of materials to Japan. Excavation Permits: No. 2016/51 (issued on 24 July 2016), No. 2017/44 (issued on 16 August 2017), No. 2018/18 (issued on 29 May 2018), No. 2019/49 (issued on 15 August 2019), and No. 2022/43 (issued on 21 August 2022). Export Permits: No. 12/5/2899 (issued on 15 August 2016), No. 12/5/3414 (issued on 18 September 2017), No. 12/5/2290 (issued on 21 June 2018), No. 12/5/382 (issued on 16 September 2019), No. 12/5/35/7 (issued on 13 September, 2022).

The western Hisma Basin in the southern Jordan area was originally investigated by D. O. Henry between 1976 and 1999[28,29,59,82,83], and the renewed fieldwork has been ongoing since 2016[84]. The eight lithic assemblages analyzed in this study are the collections of the renewed investigations. They consist of two Late Middle Paleolithic (LMP) assemblages from Tor Sabiha[29] and Tor Faraj[59], two Initial Upper Paleolithic (IUP) assemblages from Wadi Aghar[45] and Tor Fawaz[42], two Early Upper Paleolithic (EUP) assemblages from Tor Hamar Layers H and Layers F–G[62,84], and two Epipaleolithic (Epipal) assemblages from Tor Hamar Layer E2 and Layers B–E1[84–86].

The use of these materials is particularly suitable for the purpose of this study because the sites are located close to each other (within 2 km) except for Tor Sabiha, which is still only 14 km away and situated in very similar environmental settings particularly in terms of the availability of lithic raw material, chert in this case. The five sites share the same geological settings characterized by extensive exposure of Umm 'Ishrin Sandstone[87]. A few small spots of chert sources are located 2–8 km away from the Jebel Qalkha area[30], and more extensive chert outcrops are distributed in the Ma'an Plateau to the north, which is 15 km away from Jebel Qalkha and 6 km from Tor Sabiha.

**Tor Faraj.** Tor Faraj (29°56'19.9"N, 35°19'33.6"E) is a rockshelter site in the Jebel Qalkha area (Supplementary Fig. 1). It has at least 1.5 m-thick deposits, from which LMP remains have been recovered. The first investigation in 1983 opened a test trench, and more extensive excavations in 1993 and 1994 revealed an area of 7 m × 12 m[29,59]. These original excavations detected six stratigraphic layers, including Layers A, B, C, D1, D2, and E, in a stratigraphic order from the top. Layers A and B are sandy deposits of 20–30 cm in thickness that were accumulated by modern local inhabitants who leveled the shelter's floor. They are underlain by Layer C, which measures 30–75 cm in thickness and consists of homogeneous reddish-yellow silty sand. Layer D1 represents rockfall (10–35 cm in thickness) and consists of weathered sandstone rubble that is partially cemented. Layer D1 is distributed only in the southwestern corner of the excavation area. Underlying Layers D1 and C, Layer D2 consists of light red to red sand and measures 30–75 cm in thickness. The lowest layer detected so far is Layer E, which consists of reddish-yellow silty sand. The stratigraphic boundaries between Layers C, D2, and E are diffuse and gradual.

The above investigations by D. O. Henry detected three occupational levels (reported as Floors 1 and 2) in Layers C and D2 upper through intrasite spatial analyses[59]. Several radiometric dates (TL, AAR, and U-series) were obtained from Layer C, ranging between 43.8 and 69.0 kya[59].

The renewed investigation in 2017 re-opened Units A4, B2, B3, and B4[84]. Unit A4 retained deposits of Layers D2 and E while only Layer E remained in the other units. As reported by Kadowaki and Henry[84], the density of lithic artifacts in Layer E is comparable to those of Floors 1 and 2 in Layers C and D2 upper. In addition, the deposits in Layer E contained ash patches and many charcoal fragments that probably represent another occupational level. All the units yielded LMP artifacts[62], which are used in this study.

**Tor Sabiha.** Tor Sabiha (29°57'46.36"N, 35°28'10.30"E) is a rock shelter site located in the Judayid Basin, southern Jordan (Supplementary Fig. 2). The initial excavation by D. O. Henry took place in 1979–1980[29]. An area immediately outside the rock shelter was mainly excavated in a 3 m × 4 m trench, and a deposit of 1.3 m in thickness was detected. The top two layers (Layers A and B) consist of loose, reddish-brown sand and contain both Chalcolithic and LMP artifacts. An underlying Layer C consists of friable, pinkish-gray sand and contains LMP artifacts. The lowermost Layer D is white sand, lying on top of bedrock. In addition to lithic artifacts, some faunal remains were recovered, including gazelle, bos, equid, and ostrich eggshell fragments. A single AAR date (69,000 ± 6000 BP) was obtained from an ostrich eggshell fragment from Layer C[29].

In the 2019 and 2022 seasons, a renewed excavation by the authors opened eight new excavation areas (Units 100–107, each of which is 1 m × 1 m) (Supplementary Fig. 2). Five of them were placed in

the terrace next to the previous excavation areas while three units (Units 105–107) were opened inside the rock shelter. In the terrace (Units 100–102 and 104), the excavation revealed about 1 m-thick sand deposits, in which LMP artifacts were mainly collected from Layers C and D. Inside the rock shelter (Units 105–107), the excavation uncovered about 4 m-thick deposits on top of the bedrock. While most of the deposits are silty and date to the Holocene with almost no artifacts, LMP lithic artifacts and some faunal remains suddenly appear in the lowest layer (Layer 11). This layer is about 60 cm in thickness and composed of friable, pinkish-gray sand, likely corresponding to Layer C on the terrace. This study uses LMP lithic artifacts collected from Layers C, D, and 11 in the renewed investigation.

**Wadi Aghar.** Wadi Aghar is a shallow rock shelter site (29°56'11.99"N, 35°19'53.53"E) in the Jebel Qalkha area (Supplementary Fig. 3). The site was initially investigated in the 1983–1984 season[88], in which several 1 × 1 m squares were opened within a surface scatter of lithic artifacts between the bedrock wall and the large boulder. The renewed investigation in the 2016 and 2018 seasons excavated a few square meters besides the previous excavation areas[45].

The deposits are less than 1 m in thickness, within which Layers A, B, C, D1, and D2 were detected. The previous work by D. O. Henry excavated Layers A–C, while the renewed work excavated Layers B–D2. Layer A consists of powdery, grayish-tan sand, and Layer B contains light, reddish-brown sandy silt. Layer C consists of cemented sand, and it is underlain by orange, sandy deposits of Layer D. The upper part of Layer D (Layer D1) is less compact and contains many lithic artifacts, while it becomes increasingly compact in the lower part (Layer D2) with only a few lithics.

The lithic assemblages from the two investigations are techno-typologically similar to each other, and both can be affiliated with IUP[45,62,88]. This study uses the IUP assemblage from Layers C–D1 that were dated to 45–40 ka by Optically Stimulated Luminescence (OSL) and radiocarbon dating[45].

**Tor Fawaz.** Tor Fawaz is another rock shelter site (29°56'49.44"N, 35°20'9.03" E) located in the Jebel Qalkha area (Supplementary Fig. 4). The initial excavation by D. O. Henry in 1983/84 opened five 1 m × 1 m units[29], and the following investigation in 1994 excavated a larger area (3 m × 4 m) within the rock shelter where about 1 m-thick deposits were divided into Layers A, B1, B2, C, and D from the top[53]. Layer A is a recent deposit near the surface, consisting of loose, dark gray silt. Layer B consists of silty deposits. The lower part of Layer B deposits is compacted and underlain by Layer C, which consists of very compact yellow silt. The yellow silt of Layer C is partly underlain by red sand (Layer D) resting on bedrock.

The renewed excavation in 2017 set up five 1 m × 1 m units (Units 6–10). Units 6 and 10 were excavated to the depth of 30–45 cm below the surface, while only surface finds were collected in Units 7–9[42]. The excavation of Units 6 and 10 detected Layers B and C, from which more than 5000 lithic artifacts were recovered. We suggested that most of the lithic artifacts were re-reposited on the basis of multiple examinations, including OSL dating, radiocarbon dating, micromorphology, and the analysis of phytolith and dung spherulite. However, the lithic assemblages consistently show IUP techno-morphological characteristics despite a few possible inclusions of Ahmarian artifacts in 1994 trench[42,62]. Considering the complicated depositional processes, we estimated the dates of IUP occupations around ca. 45–36 ka[42]. This study used the lithic collections from Units 6a, 6b, and 10a, where no inclusion of later lithic artifacts was detected.

**Tor Hamar.** Tor Hamar (29°56'17.34"N, 35°19'8.90"E) is a rockshelter site in the Jebel Qalkha area (Supplementary Fig. 5). The initial sounding in 1983/84 opened two 1 m × 1 m test pits (Units 1 and 2), and a more substantial excavation in 1988 opened two 2 m × 2 m squares

(Units 3–10)[88]. The renewed investigations since 2016 continued excavation in Units 7–10 and newly opened Unit 11.

The excavations revealed more than 2 m-thick deposits that were divided into 11 layers: Layers A, B, C, D, E1, E2, F, G upper, G lower, H, and I in a stratigraphic order from the top. Of these layers, Layers A–E2 constitute Epipaleolithic (Epipal) cultural deposits, and they can be subdivided into the Middle Epipal (Mushabian) component in Layers A–E1 and the Early Epipal (Qalkhan or Nebekian) component in Layer E2. The lower layers (Layers F–H) yielded EUP bladelet assemblages.

Layer A is a topsoil consisting of soft yellow-red fine sand. It is likely a surface exposure of underlying Layers B–D that also consist of yellow-red fine sand but are more compact and include ash concentrations. This deposit is thicker in Units 3–6, where it was subdivided into Layers B, C, and D. In Layer C, a hearth was found in association with a probable windbreak feature and a grinding stone[29]. Layers B–D become thinner towards downslope and cannot be separated from each other in Units 7–10. The lower part of Layer D contains a large amount of sandstone rubble that is underlain by dark gray deposits with ash and charcoal (Layer E1). Layer E1 is underlain by light brown sandy silt deposits (Layer E2) in Units 9, 10, and partly 11. Layer E2 contains much less ash and charcoal than Layer E1. The northern stratigraphic section of Unit 11 shows that Layer E2 is obliquely cut by Layer E1 and not detectable in Units 3–6.

Lithic artifacts from Layers A–E2 include many microliths, and their production uses the microburin technique. Microliths from Layers A–E1 are characterized by the abundance of arch-backed bladelets, straight-backed points, and La Mouillah points. Blades/bladelets of Layers A–E1 are slightly broader than those of Layer E2. These techno-morphological characteristics are consistent between the previous and new collections, and they are affiliated with the Mushabian industry that belongs to the Middle Epipal[84,89,90] In addition, faunal remains in Layers A–E1 are relatively well preserved[85,86,91], and they also characteristically include abundant marine shells, many of which are perforated[92]. A series of radiocarbon dates indicate 15.5–15.2 ka cal. BP for the Mushabian occupations[85].

Microliths from Layer E2 are characterized by the abundance of narrow arch-backed and pointed bladelets or "attenuated lunates"[90]. Blades/bladelets of Layer E2 are narrower than those of Layer A–E1. Although the previous collections included Qalkhan points as a diagnostic type of the Qalkhan industry[29], this type was not recognized in the new collection. Instead, we identified several scalene bladelets, which might be called Qalkhan points, but anyhow, their frequency is low. Thus, the lithic assemblage from the renewed excavation is virtually indistinguishable from the Nebekian industry. Regardless of this cultural taxonomic issue, the Layer E2 assemblage is stratigraphically lower than the Mushabian and associated with radiocarbon dates around 24–18 ka cal. BP[85], which is consistent with the designation of the Early Epipal.

In contrast to the Epipal deposits (Layers A–E2), the underlying EUP layers (F–I) are distributed more continuously across all the excavation areas. Layers F and G consist of light brown sandy silt like Layer E2, and the boundary between Layer E2 and Layer F is unclear. The upper part of Layer G is ashy while it becomes progressively compact towards the lower part. The bottom of Layer G is a distinct stratum of very compact deposits of angular sandstone rubble. This gravel layer (Layer G lower) is underlain by Layer H, which consists of compact red sand. Layer H is further underlain by a very compact, light brown silt with rubble (Layer I).

Lithic technology from Layers F–H is characterized by the production of bladelets with small butts and fine overhang removals. These techno-morphological features separate the EUP from the IUP, and they appear in the Ahmarian industry[62]. Several el-Wad points were recovered from Layers F and G, and thus, the assemblages from these layers were grouped together as the Ahmarian assemblage. In addition to these techno-typological characteristics, the dominance of

unidirectional flaking is consistent with the previous collections[88], indicating the southern Ahmarian affiliation[93]. A single radiocarbon date from Layer F is 38–37 ka cal. BP[85].

Lithic artifacts from Layer H are also characterized by bladelet production but separated from Layer G because of clear stratigraphic differences marked by red sand in Layer H. In the Jebel Qalkha area, Ahmarian assemblages are often deposited in yellow silt, as attested at Tor Hamar Layers F–G, Tor Aeid, and Jebel Humeima[94]. Thus, the occurrence of bladelet assemblage in red sand is unique to Layer H of Tor Hamar. Although this might indicate chronological/environmental differences, such possibilities are currently under study.

## Measurement of cutting-edge length

To make accurate and precise measurements of irregular edge forms, we measured the length of the outlines of lithic blanks from their digital photographs by following the methods proposed by Eren et al.[18] and followed by other researchers[20,95] (Supplementary Fig. 7a). Lithic blanks were placed with their ventral face down on a sheet of graph paper, and their digital photographs were taken. The digital photos were then processed in Adobe Photoshop versions 17–23 to adjust the scale and to clarify the edges, which were then automatically traced by Adobe Illustrator versions 20–27 to extract outlines. The length of the outlines was measured in Adobe Illustrator. Following the instructions in Eren et al.[18] and Muller and Clarkson[20], we made close observations of each stone tool to exclude dull edges, such as the striking platform, broken edges, and obtuse angles, from the measurement.

Retouched tools were excluded from the analysis because of difficulty in estimating the original length of their cutting edge[22]. We also did not measure the length of the cutting edge of chips, which are flakes whose maximum length is smaller than 25 mm[18]. Microburins were also excluded from the edge-length measurement as they are tiny byproducts in the manufacturing process of microliths. Cores and chunks were also excluded as they do not have sharp cutting edges. Chunks are fragmented pieces that do not clearly show ventral or dorsal surfaces.

Thus, we measured the cutting-edge length of unmodified blanks, which consist of several techno-morphological types, such as flakes, blades, bladelets, and core trimming elements (CTE). A blade is defined as a flake whose length is equal to or greater than twice its width. Usually, a blade also has parallel lateral sides and ridges. We follow a definition of bladelet proposed by Tixier[96]. A bladelet is a blade with a length of <50 mm and a width of <12 mm. CTEs are flakes or blades with distinctive morphologies, such as core tablets and crested pieces, which result from their detachment from specific parts of cores for the maintenance of core-surface morphology. Crested blades and core tablets follow the definitions by Inizan et al.[16].

The use of unmodified blanks, as described above, is a standard method for calculating the production rate of cutting-edge. However, when we interpret the results in terms of technological behaviors, it is important to be aware that not all the unmodified blanks were actually used or intended for use. Although it would be ideal for extracting "end products" desired by knappers, such classification is difficult to realize objectively as it is widely known that apparent "by-products", such as CTEs, are often retouched into tools like scrapers.

## Calculation of cutting-edge production rates

The production rate of lithic cutting-edge is basically quantified by a ratio of the cutting-edge length to the mass of stone tools. In this study, we employed two types of ratios (Supplementary Fig. 7b).

The first type, called "the total length/mass ratio", uses the sum of the cutting-edge length of all unmodified blanks in the lithic assemblage and then divides it by the total mass of all unmodified blanks, cores, chunks, chips, and microburins. This ratio represents the total production rate of cutting-edge from the entire mass of lithic raw material, and it is theoretically close to the ratio of cutting-edge to core

mass (CE:CM) in Eren et al.[18] and the cutting edge per gram of core (mm/g) in Muller and Clarkson[20]. A single value of the total length/mass ratio is obtained for each of the lithic assemblages.

The second type, called "the individual length/mass ratio", uses the cutting-edge length of each of the unmodified blanks and then divide it by the mass of each blank. Thus, this ratio is obtained for each of the unmodified blanks that constitute a lithic assemblage, and a group of the ratios characterizes the cutting-edge production efficiency of the lithic assemblage. In contrast to the total length/mass ratio, the individual length/mass ratio does not include the mass of cores, chunks, chips, and microburins in its calculation.

## Statistical test

We used Microsoft Excel 2019 to organize the data of cutting-edge length, mass, and several morphometric measurements of the lithic artifacts analyzed in this study. Because most of the measurement data in this study deviate from the normal distribution, we used the non-parametric Kruskal–Wallis test and the post hoc Dunn's test for multiple comparisons with Bonferroni correction to examine whether differences among the lithic assemblages were significant in terms of the cutting-edge production rates and lithic morpho-metric measurements, such as length, width, thickness, and the platform size. All tests were two-tailed where applicable. All box plots indicate the median (middle line) and interquartile range (box). The upper and lower ends of the whiskers represent the maximum and minimum, excluding outliers (single points) defined by the Tukey method. The Spearman correlation coefficient was obtained to examine the significance of correlations between the cutting-edge production rate and lithic morpho-metric measurements. These statistical examinations were performed with IBM SPSS version 27.

## Reporting summary

Further information on research design is available in the Nature Portfolio Reporting Summary linked to this article.

## Data availability

The data of cutting-edge length, mass, and several morphometric measurements of the Paleolithic stone tools analyzed in this study are available to the public in the figshare. https://doi.org/10.6084/m9.figshare.23577093. The lithic assemblages analyzed in this study are stored in the Nagoya University Museum, Japan, and access to the materials can be arranged by the corresponding author (S. Kadowaki). Source data are provided as a Source Data file. Source data are provided in this paper.

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

## Acknowledgements

This research derives from a joint project, "Cultural History of PaleoAsia" directed by Yoshihiro Nishiaki (The University of Tokyo). S.K. acknowledges support from MEXT KAKENHI (No. 16H06409, 20H00026, 23K17275), JPJSBP (No. 120228403), the Mitsubishi Foundation (No. 30205), and the Suntory Foundation (No. 2023-111). This research was also funded by JSPS Fellows (No. 22KJ1518) to E.S. and the Murata Science Foundation (No. M23-026) to M.H. The authors are deeply grateful to the late Prof. Donald O. Henry for his warm support and encouragement in the renewed investigations of prehistoric sites in southern Jordan. This paper is dedicated to him as a tribute to honor his enduring legacy. We also appreciate permission to conduct fieldwork in southern Jordan and generous support from Fadi Balawai (Director General), Aktham Oweidi, Maher Amreen, and other staff members of the Department of Antiquities of Jordan. The fieldwork in Jordan was accomplished by cooperation from crew members (Hidekazu Yoshida, Hitoshi Hasegawa, Dustin White, Shoji Nishimoto, Yuichi Naito, Ryosuke Yamauchi, Kazunobu Ikeya, Momoko Osawa, Ahmad Thaher, Jianjie Yin, and Nanako Kimoto) and Humayma community members (Adel Mohammad, Ahmad Sabah, Ahmad Suleiman Al Husaseen, Ali Mohamad, Awdi Jzeelat, Fadi Jzeelat, Faraj Mohammad, Faraj Salim, Halid Jzeelat, Ibraheem Faraj Abu Zifoon, Ibraheem Salem Analeen, Ibrahim Jzeelat, Joma Mohammad Arjaramyah, Mohammad Faraj, Mohammad Jzeelat, Mutab Saleem Jutheelat, Salim Mohamad, and Thamer Sleiman). We also thank Risako Kida for her help in OSL dating and Miki Sano for her support in the lithic analysis.

## Author contributions

S.K. designed and directed the project. S.M., O.T., M.H., E.S., K.T., T.T. and J.Y.W. contributed to the fieldwork in Jordan and the collection of the lithic samples. T.T. conducted chronological analyses of the archeological sites. S.K. conducted the lithic techno-morphological analyses. K.T. and E.S. contributed to the lithic analyses. A.W., S.K. and M.H. performed the measurements of lithic edge length. J.Y.W. contributed to the statistical analyses. S.K. took the lead in writing the paper. All authors provided feedback and helped shape the paper.

## Competing interests

The authors declare no competing interests.
