## [Peer Review File · Nature Communications]

Delayed increase in stone tool cutting-edge productivity at the Middle-Upper Paleolithic transition in southern JordanReviewers' Comments:

Reviewer #1:

Remarks to the Author:

I think that the authors have done an excellent job at addressing the concerns and comments of the reviewers. Reviewers will always complain about the number of sites and the sample sizes (particularly those who did not actually do real comparative studies it seems!). I think the authors have a good and representative sample here, and make good arguments along these lines. In my opinion, the paper is ready to be published. I just have a few very minor points:

I think the abstract needs a sentence or so at the end summarising the wider implications of the findings.

Line 24, "more recent increase" is slightly odd grammar, better to say something like "Recently there has been an expansion of the archaeological record" or "improved knowledge of the archaeological records in Africa and Asia" etc

Line 35, "Most of" should be just "Most"

Line 234, should be "the behavioral significance" not just "behavioral significance"

Line 381....this is a true enough point, but given that another possibility is that IUP represents a new group coming from Africa...then the point is the background here is also things like platform faceting, hard hammer percussion etc....these are not distinctive local features which imply continuity. So I would emphasise more continuity of broad patterns of behaviour rather than specific continuity of technological methods and or actual social groups.

Line 382 – should be "the long-term" not "a long-term"

Line 394, should be "permission to conduct fieldwork" not "permission of fieldwork"

Given that an important outcome of the study is an emphasis on the importance of miniaturization...I suggest the addition of a reference at some suitable point, Shipton et al. (2018) in Nature Communications on the site of Panga ya Saidi. This is a long MSA-LSA cultural sequence. The key point here is that within the MIS 4-2 deposits every single different way that the LSA has been defined comes and goes...the one thing that stays consistent is that there is a switch to miniaturization around 71 ka. This then is key technological change in this sequence. I just think this a useful analogous point which is worth mentioning in the discussion perhaps.

Reviewer #2:

Remarks to the Author:

In this study, the authors examined a series of Paleolithic assemblages from Jordan to evaluate changes in lithic cutting edge production rate over time. Using the ratio of cutting edge to mass in individual flakes and whole assemblages as a proxy for the efficiency of cutting edge production, the study showed that increases in the rate of cutting edge production did not correspond with the Middle/Upper Paleolithic transition, and instead appear to relate to the development of bladelet technology in the Early Upper Paleolithic. Interestingly, the IUP assemblages produced the lowest cutting-edge productivity rates, which the authors attributed to the manufacture of large blades that would have been suitable for prolonged use.

There are several noteworthy aspects of this study that have significant implications on Paleolithic archaeology. First, this is among the first systematic analysis using objective quantitative measures to

examine diachronic patterns in Paleolithic technology in the study region. The methodology is sound and clearly described. The large sample size of the artifacts analysed adds valuable new data to Paleolithic research. I strongly encourage the authors to make their data openly available at some point to allow future studies to conduct more larger scale, big data inter-assembly analyses. Second, the results provide important empirical evidence highlighting variability and complexity of Paleolithic assemblages that are incompatible with conventional linear models of Paleolithic cultural evolution.

I think the authors have done a good job addressing the concerns raised in the previous round of reviews. In particular, shifting the paper's focus away from broad evolutionary processes to specific technological developments has made the revised paper stronger and more substantiated. However, I find the current justification for the study in the Introduction a bit weak, and I think the authors can make a stronger and more compelling case for the significance of this study. Specifically, in lines 91-94, the authors noted that previous studies have examined cutting edge production patterns across the MP and the UP in the Levant, and these studies detected a slight increase. The authors argued that these results need further clarification for three reasons - 1) small sample size, 2) the lack of IUP in the sample, and 3) imprecise methods. The first and third reasons are self-explanatory, but the second reason regarding the IUP should be elaborated upon. What is the IUP and why is it important? If I were the author, I would emphasize that IUP is an extremely important phenomenon, particularly in this region. Its absence in previous studies represents a significant knowledge gap.

My other comments are relatively minor, some of which echo comments from Reviewer #2 in the previous round of review regarding the assumption of temporal sequence among the assemblages. I see that the authors have specified the grouping of the assemblages and clarified in figure captions regarding their assumed temporal relationships (or lack thereof). However, the way these figures are presented can still be confusing to readers. Take Figure 2 as an example, while the authors have clarified in the caption that no chronological sequence is assumed between LMP1 and LMP2 and between IUP1 and IUP2, the lines still imply a sequence. This is confusing because a temporal sequence is indeed represented for the later assemblages. Thus, despite the clarification in the caption, it is still very easy for readers to misinterpret the figure. The same comment can be extended to Figure 3a and 6. I recommend the authors to combine LMP1 and LMP2 into a single assemblage unit called 'LMP', and IUP1 and IUP2 into a single 'IUP' unit. Looking at the data presented, combining these assemblages is unlikely to change the general results of this study (hence not impact the conclusion), but these combined assemblages would be not only clear and easier to understand, but also consistent with the assemblage groupings defined in lines 88-90.

Other minor comments:

- I do not understand the purpose of Figure 4. The relationship between edge length per mass and flake width is barely mentioned in the paper. I can see the point that cutting edge length ratio decreases with flake size - if this is the point, it needs to be better fleshed out in the results section.
- Line 24-25: "However, more recent increase in archaeological records..." The wording here is odd. The archaeological record is typically defined as the body of physical evidence left behind by past human activities. I assume the authors meant something more along the line of "more recent increase in the recovery of archaeological records...".
- Line 67: Consider citing the study by Hoggard and Stade (2019) on the comparison of cutting edge production between Levallois and UP blade technologies. Hoggard, C. S., & Stade, C. M. (2019). The efficiency of Middle Palaeolithic technological blade strategies: An experimental investigation. *Lithics—The Journal of the Lithic Studies Society*, (39), 52.

Responses to the reviewer's comments

We appreciate the reviewers' careful reading and comments to our revised manuscript. The following shows our responses to the comments.

Reviewer #1 (Remarks to the Author):

I think that the authors have done an excellent job at addressing the concerns and comments of the reviewers. Reviewers will always complain about the number of sites and the sample sizes (particularly those who did not actually do real comparative studies it seems!). I think the authors have a good and representative sample here, and make good arguments along these lines. In my opinion, the paper is ready to be published. I just have a few very minor points:

Reply

We deeply appreciate the reviewer's positive comments to the revision of our paper.

I think the abstract needs a sentence or so at the end summarising the wider implications of the findings.

Reply

Thank you for the suggestion. We added a sentence at the end of the abstract to highlight the wider implications of our study. In addition, we added a sentence to provide the background of the study according to the formatting instructions of Nature Communications.

Line 24, "more recent increase" is slightly odd grammar, better to say something like "Recently there has been an expansion of the archaeological record" or "improved knowledge of the archaeological records in Africa and Asia" etc

Reply

Thank you for the comments and suggestion of re-phrasing. This correction was also pointed out by Reviewer #2. We corrected a sentence accordingly. See Line 23 in the revised text.

Line 35, "Most of" should be just "Most"

Reply

Corrected. See Line 35 in the revised text.

Line 234, should be "the behavioral significance" not just "behavioral significance"

Reply

Corrected. See Line 174 in the revised text.

Line 381....this is a true enough point, but given that another possibility is that IUP represents a new

group coming from Africa...then the point is the background here is also things like platform faceting, hard hammer percussion etc....these are not distinctive local features which imply continuity. So I would emphasise more continuity of broad patterns of behaviour rather than specific continuity of technological methods and or actual social groups.

Reply

Thank you for the thoughtful comments about the recognition of “cultural continuity”. We agree that some lithic production techniques that were prevalent in the MP, such as platform faceting, could occur as technological convergence and may not be strong evidence for the continuity of specific cultures or social groups. To clarify our argument, we deleted a word “continuity” from the sentences. See Lines 275 and 281 in the revised text.

Line 382 – should be “the long-term” not “a long-term”

Reply

Corrected. See Line 278 in the revised text.

Line 394, should be “permission to conduct fieldwork” not “permission of fieldwork”

Reply

Corrected.

Given that an important outcome of the study is an emphasis on the importance of miniaturization...I suggest the addition of a reference at some suitable point, Shipton et al. (2018) in Nature Communications on the site of Panga ya Saidi. This is a long MSA-LSA cultural sequence. The key point here is that within the MIS 4-2 deposits every single different way that the LSA has been defined comes and goes...the one thing that stays consistent is that there is a switch to miniaturization around 71 ka. This then is key technological change in this sequence. I just think this a useful analogous point which is worth mentioning in the discussion perhap.

Reply

Thank you for suggesting a broad implication of our study. We agree that our study of the link between the increased edge productivity in the EUP and the lithic miniaturization is relevant to the discussion on the lithic miniaturization and its behavioral implications at the MSA-LSA transition in Africa. Thus, we added the citation of Shipton et al. (2018) in Nature Communications, and we also added their subsequent, more recent study (Shipton et al. 2021 in Journal of Human Evolution) that focuses on the lithic technology. See Line 261-264 in the revised text.

Reviewer #2 (Remarks to the Author):

In this study, the authors examined a series of Paleolithic assemblages from Jordan to evaluate changes in lithic cutting edge production rate over time. Using the ratio of cutting edge to mass in individual flakes and whole assemblages as a proxy for the efficiency of cutting edge production, the study showed that increases in the rate of cutting edge production did not correspond with the Middle/Upper Paleolithic transition, and instead appear to relate to the development of bladelet technology in the Early Upper Paleolithic. Interestingly, the IUP assemblages produced the lowest cutting-edge productivity rates, which the authors attributed to the manufacture of large blades that would have been suitable for prolonged use.

There are several noteworthy aspects of this study that have significant implications on Paleolithic archaeology. First, this is among the first systematic analysis using objective quantitative measures to examine diachronic patterns in Paleolithic technology in the study region. The methodology is sound and clearly described. The large sample size of the artifacts analysed adds valuable new data to Paleolithic research. I strongly encourage the authors to make their data openly available at some point to allow future studies to conduct more larger scale, big data inter-assemblage analyses. Second, the results provide important empirical evidence highlighting variability and complexity of Paleolithic assemblages that are incompatible with conventional linear models of Paleolithic cultural evolution.

I think the authors have done a good job addressing the concerns raised in the previous round of reviews. In particular, shifting the paper's focus away from broad evolutionary processes to specific technological developments has made the revised paper stronger and more substantiated.

Reply

We appreciate the reviewer's careful following of the revision process. Regarding the data of this study, we will make them openly available. Below is our statement in Data Availability.

The data of cutting-edge length, mass, and several morphometric measurements of Paleolithic stone tools analyzed in this study are available to the public in the figshare. <https://doi.org/10.6084/m9.figshare.23577093>

However, I find the current justification for the study in the Introduction a bit weak, and I think the authors can make a stronger and more compelling case for the significance of this study. Specifically, in lines 91-94, the authors noted that previous studies have examined cutting edge production patterns across the MP and the UP in the Levant, and these studies detected a slight increase. The authors argued that these results need further clarification for three reasons - 1) small sample size, 2) the lack of IUP in the sample, and 3) imprecise methods. The first and third reasons are self-explanatory, but the second reason regarding the IUP should be elaborated upon. What is the IUP and why is it important? If I were the author, I would emphasize that IUP is an extremely important phenomenon, particularly in this region. Its absence in previous studies represents a significant knowledge gap.

Reply

We appreciate the reviewer's suggestion to emphasize the significance of the IUP samples included in this study. Following the comments, we added a few sentences to explain the significance of the IUP chrono-cultural concept by referring to its wide geographic implications and its association with the dispersals of *Homo sapiens*.

See Lines 87-91 in the revised text.

My other comments are relatively minor, some of which echo comments from Reviewer #2 in the previous round of review regarding the assumption of temporal sequence among the assemblages. I see that the authors have specified the grouping of the assemblages and clarified in figure captions regarding their assumed temporal relationships (or lack thereof). However, the way these figures are presented can still be confusing to readers. Take Figure 2 as an example, while the authors have clarified in the caption that no chronological sequence is assumed between LMP1 and LMP2 and between IUP1 and IUP2, the lines still imply a sequence. This is confusing because a temporal sequence is indeed represented for the later assemblages. Thus, despite the clarification in the caption, it is still very easy for readers to misinterpret the figure. The same comment can be extended to Figure 3a and 6. I recommend the authors to combine LMP1 and LMP2 into a single assemblage unit called 'LMP', and IUP1 and IUP2 into a single 'IUP' unit. Looking at the data presented, combining these assemblages is unlikely to change the general results of this study (hence not impact the conclusion), but these combined assemblages would be not only clear and easier to understand, but also consistent with the assemblage groupings defined in lines 88-90.

Reply

Thank you for the suggestion to make further clarification of the chrono-cultural units for comparison in this study. As the reviewer points out, we do not assume the chronological order between LMP1 and LMP2, but these data are taken from different sites. Thus, we consider them as samples from different distributions and would like to keep our current statistical results. The same argument applies for IUP1 and IUP2.

However, we understand that the numerical order of the two assemblages in the LMP and IUP (i.e., LMP1, LMP2, IUP1, and IUP2) and the layout of the graphs can be still misleading. So, we removed the use of "LMP1, LMP2, IUP1, and IUP2" and added dotted lines in the graphs to clarify the chrono-cultural units used in this study. This led to the revision of most of the figures and the table.

Other minor comments:

- I do not understand the purpose of Figure 4. The relationship between edge length per mass and flake width is barely mentioned in the paper. I can see the point that cutting edge length ratio decreases with

flake size - if this is the point, it needs to be better fleshed out in the results section.

Reply

The suggested point is exactly our purpose of Figure 4. Thus, we added the following explanation. See Line 130-132 in the revised text.

“This means that narrower and thinner blanks with smaller striking platforms tend to have the greater ratios of the edge length per mass. As typically shown in the case of width (Fig. 4), such a negative correlation is similarly observable in each of the debitage types.”

- Line 24-25: "However, more recent increase in archaeological records..." The wording here is odd. The archaeological record is typically defined as the body of physical evidence left behind by past human activities. I assume the authors meant something more along the line of "more recent increase in the recovery of archaeological records..."?

Reply

Thank you for the comments and suggestion of re-phrasing. This correction was also pointed out by Reviewer #1. We corrected a sentence accordingly. See Line 23 in the revised text.

- Line 67: Consider citing the study by Hoggard and Stade (2019) on the comparison of cutting edge production between Levallois and UP blade technologies. Hoggard, C. S., & Stade, C. M. (2019). The efficiency of Middle Palaeolithic technological blade strategies: An experimental investigation. *Lithics—The Journal of the Lithic Studies Society*, (39), 52.

Reply

We were not aware of this study and highly appreciate the suggestion to add it in the citations. We cited it in the Methods. See Line 295 in the revised text.